# The Radiographic Risk Factors Predicting Occurrence of Idiopathic Carpal Tunnel Syndrome in Simple Wrist X-ray

**DOI:** 10.3390/jcm11144031

**Published:** 2022-07-12

**Authors:** Beom-Su Han, Ki-Hong Kim, Kyu-Jin Kim

**Affiliations:** Department of Orthopedic Surgery, Seoul Medical Center, 156, Sinnae-ro, Jungnang-gu, Seoul 02053, Korea; hans7194@naver.com (B.-S.H.); rlghd7186@naver.com (K.-H.K.)

**Keywords:** idiopathic carpal tunnel syndrome, development, radiographic, risk factors

## Abstract

The causes of carpal tunnel syndrome are complex. However, little is known about the risk factors for carpal tunnel syndrome occurrence on simple radiographic images. To determine the X-ray imaging factors that can predict idiopathic carpal tunnel syndrome occurrence, we compared a group diagnosed with idiopathic carpal tunnel syndrome who received carpal tunnel release with a control group that had no symptoms. The simple wrist X-ray findings of 75 patients diagnosed with idiopathic carpal tunnel syndrome and 87 patients selected for the control group were evaluated. All the carpal tunnel syndrome patients were diagnosed based on clinical symptoms and nerve conduction studies. Anteroposterior and lateral radiographs of the wrists were taken in all the groups. The radial inclination, volar tilt, ulnar variance, radiolunate angle, and lunate-to-axis-of-radius distance were measured. Data were measured using two independent raters. After calculating the average of each value, the two groups were statistically compared. The diagnostic performance of statistically different figures was confirmed by drawing receiver operator characteristic curves. There was a significant difference in the radiolunate angle and lunate-to-axis-of-radius distance between the two groups (*p* < 0.01 and *p* = 0.028, respectively). The odd ratios for each parameter were 1.052 and 1.319, respectively. The area under the receiver operator characteristic curves were 0.715 and 0.601, respectively. In this study, radiolunate angle and lunate-to-axis-of-radius distance were useful as radiographic diagnostic tools. In other words, excessive dorsiflexion and volar displacement of the lunate can be considered as risk factors that may cause idiopathic carpal tunnel syndrome in the future.

## 1. Introduction

Carpal tunnel syndrome (CTS) is one of the most common upper-extremity peripheral neuropathies [1], and its causes are multifactorial. Idiopathic anatomic anomalies, such as ganglion cysts, occupational mechanical stresses, and systemic diseases, including obesity, drug toxicity, alcoholism, diabetes, hypothyroidism, and rheumatoid arthritis are causes of CTS [2]. Additionally, the narrowing of the carpal tunnel is one of the well-known causes of idiopathic CTS [3,4,5]. The causes of carpal tunnel narrowing and elevation of carpal tunnel pressure are numerous, including space-occupying lesions, repetitive finger and wrist flexion and extension movements, tenosynovitis of the flexor tendons, and malunion of distal radius fractures [1,6,7].

Malunion of distal radial fracture has been frequently reported as the main cause of CTS development after trauma [8,9,10]. The abnormal pathway of the median nerve and narrowness of the carpal canal induced by malunion are the causes of median nerve compression [3]. This fact indicates that the bony morphology of the wrist joints is very important in the development of CTS. In clinical practice, plain radiography is usually used to check the bony morphology of the wrist joints because it is fast, easy and inexpensive compared to other examinations, such as ultrasonography and magnetic resonance imaging. However, the importance of plain radiography in patients with CTS is often underestimated. If radiological risk factors for idiopathic CTS development are disclosed, surgeons can predict CTS occurrence in advance. There are few reports suggesting plain radiographic risk factors for idiopathic CTS [6]. Therefore, our study aimed to reveal radiographic predictive factors for CTS development by comparing the radiological parameters of simple wrist X-rays between CTS patients and controls.

## 2. Materials and Methods

This was a case–control study (level of evidence: level III). Approval for this study was obtained from our institutional research and ethics board (IRB No. 2022-03-011). We randomly selected patients who were diagnosed with idiopathic CTS and underwent carpal tunnel release from January 2016 to December 2021. All the patients underwent wrist radiography at their first visit.

In our center, all patients who visit with symptoms such as pain in the wrist must take X-rays of the opposite wrist to compare with the symptomatic area and we measured the radiographic angle using X-rays of the opposite wrist and designated them as the control group. Medical records were reviewed to ensure that there were no symptoms in the opposite side.

The CTS group had 83 cases of 75 patients and 8 patients diagnosed as both wrist CTS (12 men and 63 women). The control group was 87 cases of 87 people (18 male, 69 female). There were no statistically significant differences in age or sex distribution between the two groups (Table 1).

The clinical diagnosis of idiopathic CTS was based on clinical symptoms. We diagnosed patients with CTS when they complained of numbness or a tingling sensation in the 1~3rd fingers and radial half of the fourth finger, and at least one of the provocative tests (wrist tinel sign or Phalen test) was positive. Additionally, patients with thenar muscle atrophy were considered to have CTS. We confirmed CTS by performing a nerve conduction study among patients who complained of CTS symptoms. We excluded those with underlying diseases, such as chronic renal failure, thyroid diseases, uncontrolled diabetes mellitus, and rheumatoid arthritis. Systemic CTS was not considered in this study, as these diseases are known to increase CTS prevalence systemically. Patients with ipsilateral upper-extremity trauma were excluded from the study, including those with anatomical anomalies or space-occupying lesions that were detected during carpal tunnel release. Anteroposterior and lateral radiographs were obtained in both groups. All the wrist radiographs were obtained during the first visit in both groups.

Radiology technologists did not know the patients’ symptoms. We measured the radial inclination, volar tilt, ulnar variance, radiolunate angle (RLA), and lunate-to-axis-of-radius distance (LARD) using the INFINITT PACS M6 image analysis software (INFINITT Healthcare Company). The radial inclination, volar tilt, and ulnar variance are the most frequently used parameters to identify wrist malalignment [11]. The angle between a line from the radial styloid tip to the ulnar aspect of the distal radius and a line perpendicular to the longitudinal axis of the radius were measured as the radial inclination. The angle between the line from the dorsal edge to the volar edge of the radius and the line perpendicular to the longitudinal axis of the radius were measured as the volar tilt. Ulnar variance was defined as the relative length of the distal articular surfaces of the radius (mid-line of the surface) and ulna (fovea). The RLA is the angle between the longitudinal axes of the radius and lunate in the lateral view. We marked negative values when the lunate was inclined to volar flexion and positive values when it was inclined to dorsiflexion. The LARD was defined as the perpendicular distance from the center of the lunate to the sagittal axis of the radius shaft (Figure 1).

We marked negative values when the center of the lunate locates the dorsal side compared to the sagittal axis of the radius shaft and positive values when it locates the volar side. Two raters (B and C) performed the measurements independently. Raters B and C were measured without information on clinical symptoms.

### Statistical Analysis

The results are expressed as mean ± standard deviation. We compared the radiological parameters between the CTS and control groups. Normal statistical distribution was confirmed using the Shapiro–Wilk test. We calculated the average of each rater’s values and compared the values of each group using an independent *t*-test. Among the radiologic parameters that showed significant differences between the two groups in the independent *t*-test, we calculated the odd ratios of each parameter using logistic regression analysis. Statistical significance was set at *p* < 0.05. We measured the diagnostic performance of each radiological parameter that showed significant differences in the *t*-test by drawing a receiver operating characteristic (ROC) curve. The area under the ROC curve (AUC) was measured and AUC > 0.5 was considered a significant diagnostic radiological tool [12]. The cut-off values on the ROC curves were defined by the highest point on the vertical axis and the point furthest to the left on the horizontal axis (upper-left corner). All the statistical analyses were performed using SPSS software (version 21.0; IBM Company, North Castle Drive, Armonk, NY, USA).

## 3. Results

The radial inclination was 22.7 ± 2.9° in the CTS patients and 23.1 ± 1.7° in the control group (*p* > 0.05). The volar tilt was 12.7 ± 4.2° in the CTS patients and 14.0 ± 4.1° in the control group (*p* > 0.05). The ulnar variance values were −1.4 ± 1.2 mm and −1.7 ± 1.3 mm, respectively (*p* > 0.05). The RLA was 5.4 ± 7.6° in the CTS patients and −1.1 ± 8.4° in the control group (*p* < 0.01). The LARD values were 4.3 ± 2.0 mm and 3.5 ± 2.3 mm, respectively (*p* = 0.028) (Table 2). The RLA and LARD showed significant differences between the two groups. Table 3 shows the odd ratios for each parameter.

The results of the ROC curve analysis for the RLA and LARD are shown in Figure 2 and Figure 3.

The AUC values for the RLA and LARD were 0.715 and 0.601, respectively. The cut-off values to discriminate individuals at risk of development of CTS from the asymptomatic control were 1.65° and 4.05 mm, respectively. The sensitivity and specificity of the RLA were 68.7% and 67.8%, respectively. The sensitivity and specificity of the LARD were 59.0% and 55.2%, respectively (Table 4).

## 4. Discussion

The etiology of idiopathic CTS varies and is multifactorial. The mechanical stress of the subsynovial connective tissue due to repetitive flexion and extension motion of the wrist and finger joints is a well-known cause of idiopathic CTS [13,14]. The congenital narrowness of the carpal tunnel has been suggested as one of the causes of idiopathic CTS [3]. Several reports have analyzed the pathogenesis of CTS following distal radius fractures and discussed predisposing risk factors [9,10,15,16]. This suggests that the bony morphology of wrist joints plays an important role in CTS occurrence.

The usefulness of simple radiographs in the diagnosis of CTS is not well known. Plain radiography is easy, quick, and inexpensive; however, plain radiographs do not show soft tissue status. Because of the impossibility of checking soft tissue status, the role of plain radiography is limited in CTS diagnosis [17,18].

For this reason, there are few reports of radiological predisposing factors related to the idiopathic CTS development. Ikeda et al. [6]. reported that significant differences in the ulnar variance were observed between CTS patients and controls, and suggested that the imbalance of radioulnar bone length is one of the risk factors to develop CTS. In our study, we investigated radial inclination, volar tilt and ulnar variance and compared them between the two groups and found no significant differences. However, the RLA and LARD showed significant differences between the two groups. Herein, the development of CTS increased when the RLA was over 1.65° of dorsiflexion, and this fact is essentially in agreement with the study of Buchberger and Aro et al. [10,17]. The chronic dorsiflexion of the lunate induces mechanical stress to the flexor tendons, which causes attritional tendon ruptures and median nerve compression in severe cases [19]. Additionally, the mechanical friction and stress to the subsynovial connective tissue of the flexor tendons is a well-known cause of idiopathic CTS [13,14]. Based on the above studies, we suggest that the chronic dorsiflexion of the lunate more than 1.65° is enough to cause idiopathic CTS.

In our study, the risk of developing CTS increased when the LARD increases above 4.05 mm. This implies that if the lunate moves significantly in the volar direction, the carpal tunnel becomes relatively narrow, thereby increasing the likelihood of CTS development. Kamihata et al. [20]. reported the rare case of CTS with flexor tendon rupture which was caused by scapholunate advanced collapse (SLAC) with the extrusion of the lunate into the carpal tunnel. Additionally, it has been reported that the main cause of CTS is the protrusion of the lunate into the carpal tunnel due to chronic dislocation or collapse following Kienböck’s disease [21,22]. Kuhnel et al. [23]. introduced the concept of CARD (Capitate-to-Axis-of-Radius Distance) and suggested that CARD should be corrected, as well as radial length, radial inclination, and volar tilt when distal radius fractures occurred. Using a similar concept, the LARD can also be considered as an interesting indicator, which surgeons should be interested in when they check wrist X-rays.

The strength of this study is that it investigates the association with CTS development by measuring RLA and LARD, as well as basic distal radial parameters (for example, radial inclination, volar tilt, ulnar variance). This suggests that RLA and LARD should be considered during surgery for distal radial fractures to prevent future CTS development. In addition, by presenting radiologic risk factors of idiopathic CTS occurrence, which have been little known, diagnostic tools have been prepared to predict the occurrence of idiopathic CTS in the future.

A limitation of this study is that all the CTS patients underwent surgery. In other words, there is a disadvantage that CTS patients who can be treated by conservative treatment are not included. Second, those included in the control group did not have symptoms of CTS at the time of radiography, but symptoms of CTS may occur in the future, but this is unknown. Finally, in the case of LARD, the AUC of the ROC curve (0.601) could not be considered to be relatively high. When the area of the ROC curve is >0.5 and ≤0.7, it is considered as a low-accuracy diagnostic tool [24]. Therefore, further large cohort studies are needed in the future.

## 5. Conclusions

Our results suggest that excessive dorsiflexion or volar displacement of the lunate compared to the radial shaft is a risk factor for CTS development. Additionally, surgeons should try to restore these parameters when performing surgery for distal radius fractures.

## Figures and Tables

**Figure 1 jcm-11-04031-f001:**
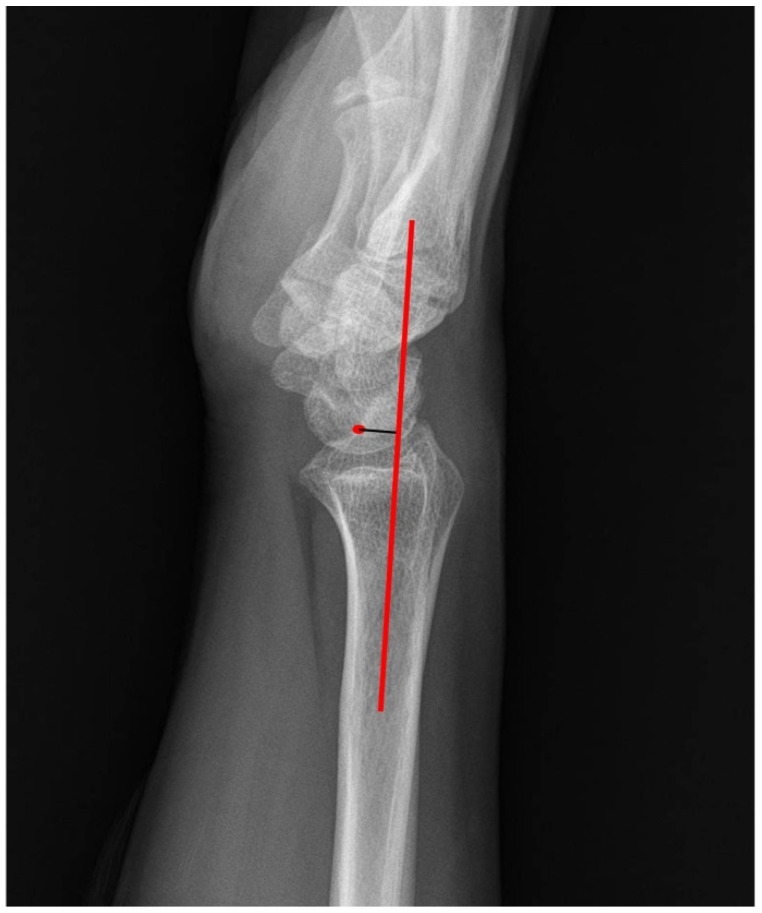
The LARD was defined as the perpendicular distance from the center of the lunate to the sagittal axis of the radius shaft.

**Figure 2 jcm-11-04031-f002:**
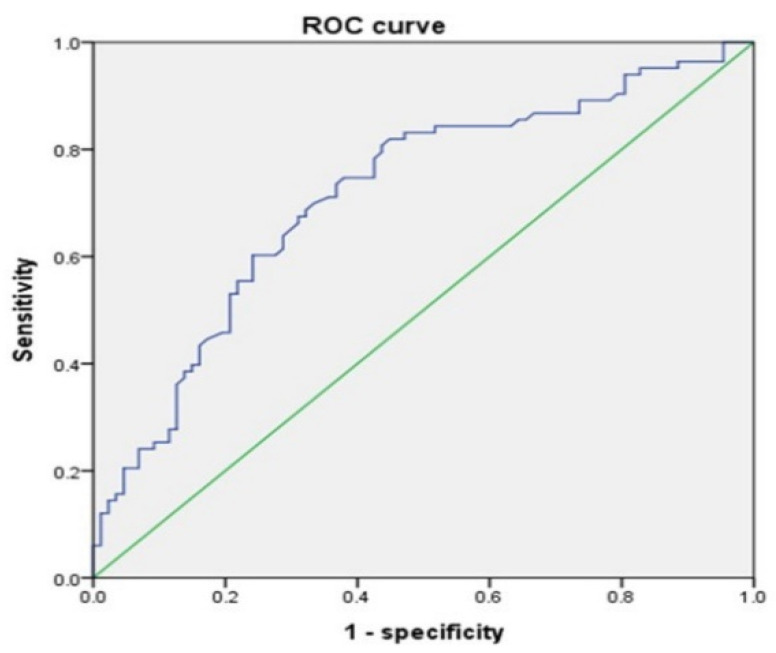
RLA ROC curve.

**Figure 3 jcm-11-04031-f003:**
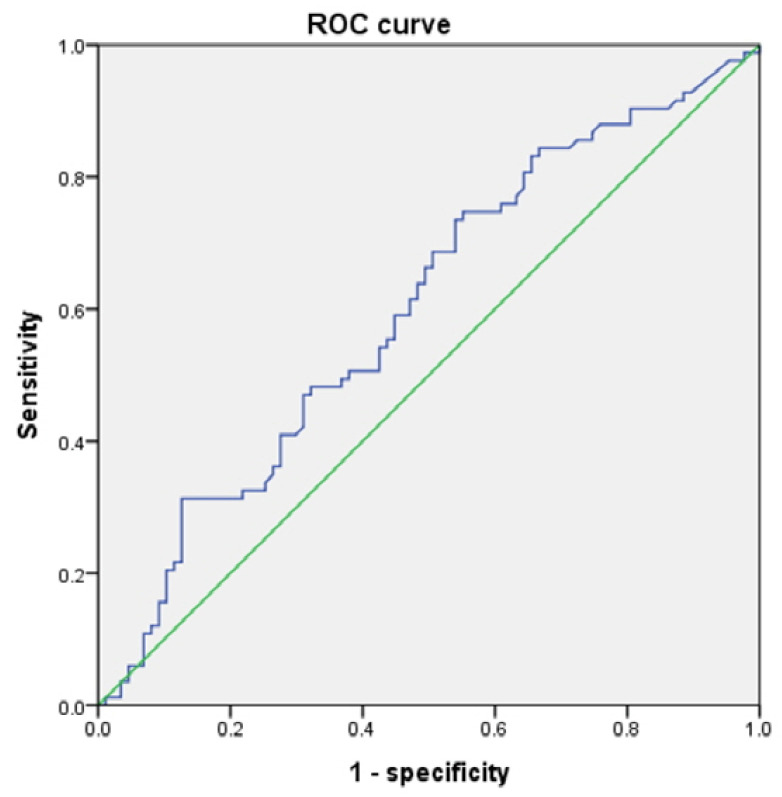
LARD ROC curve.

**Table 1 jcm-11-04031-t001:** Average age and gender of both groups.

	Age	Male	Female	Total
CTS patients	63.2 ± 11.6	12 (12 hands)	63 (71 hands)	75 (83 hands)
Control	61.9 ± 8.9	17 (17 hands)	70 (70 hands)	87 (87 hands)
*p*-value	0.41	0.22		

No statistical differences of age and sex distribution between the two groups.

**Table 2 jcm-11-04031-t002:** Results of radiological parameters of each group.

	CTS Patients	Control	*p*-Value
Radial inclination	22.7 ± 2.9°	23.1 ± 1.7°	>0.05
Volar tilt	12.7 ± 4.2°	14.0 ± 4.1°	>0.05
Ulnar variance	−1.4 ± 1.2 mm	−1.7 ± 1.3 mm	>0.05
Radiolunate angle	5.4 ± 7.6°	−1.1 ± 8.4°	<0.01 *
Lunate-to-Axis-of-Radius distance	4.3 ± 2.0 mm	3.5 ± 2.3 mm	0.028 *

* Statistically significant (*p* < 0.05).

**Table 3 jcm-11-04031-t003:** Radiologic parameters associated with development of idiopathic CTS on logistic regression analysis.

Radiologic Parameters	Odds Ratio (95% Confidence Interval)	*p*-Value
RLA	1.052 (1.007–1.1)	0.024 *
LARD	1.319 (1.085–1.605)	0.005 *

* Statistically significant (*p* < 0.05).

**Table 4 jcm-11-04031-t004:** Results of ROC curve analysis.

	AUC Values	Cut-Off Values	Sensitivity	Specificity
RLA	0.715	1.65°	68.7%	67.8%
LARD	0.601	4.05 mm	59.0%	55.2%

## Data Availability

The data presented in this study are available on request from the corresponding author. The data are not publicly available due to privacy restrictions.

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
