# Peer review of "The Radiographic Risk Factors Predicting Occurrence of Idiopathic Carpal Tunnel Syndrome in Simple Wrist X-ray"

_jcm, 2022, doi:10.3390/jcm11144031_

Round 1

Reviewer 1 Report

This paper discusses which radiographic elements are characteristic in patients with carpal tunnel syndrome (CTS) by comparing radiographic images of the wrist joint between CTS patients and non-CTS patients. The main contribution of the paper is to present which radiographic features are risk factors for idiopathic carpal tunnel syndrome.

I recommend that this paper be accepted after major revision.

Major comment:

This paper looks attractive in that it presents the radiographic features with CTS that were not previously evident. The need for this study in the Background section, the comparative study to fulfill their objectives, and the usefulness of the results obtained are all well written. However, it has questionable points that could be a fatal problem.

1. Page 3, line 104.

Based on your description, since rater A measured the radiographic factors unblinded, his/her data should have been biased.

Ideally, please include another rater (rater C) who is blinded and have him/her measure the data. Then, please remove the data measured by rater A, add the data measured by rater C, and analyze them again.

If you cannot do it for some reason, please present the data obtained by each rater and the agreement/consistency between the two raters' results. Those might allow you to indicate that the measurement values obtained by rater A did not favorably affect the analysis results.

2. Page 4, line 133 (Table 1).

In your study, in which many women were included, the RLA of the control group was -3.9 degrees, but in the following past references A and B, the RLA is presented as 9.3 and 3.3 degrees, respectively. In particular, reference B shows that RLA is significantly greater in women. Unlike your study, the studies in references A and B involved almost equal numbers of men and women; thus, I must say that there is a rather large difference between their results and yours. If these were the results in the control group in your study, it is possible that you would not have obtained a significant difference.

Then, looking at your methods section, you state that you randomly included patients as a control group who were performed x-ray for screening purposes. What disease or reason did these patients have their wrist x-rays done for screening purposes? Were they really appropriate as a control group? Please show the evidence that the data in the control group were appropriate enough for the research. Also, please discuss and provide the reason for the differences in results from previous papers.

Both above two points are critical points that allow you to arbitrarily manipulate the results of your study. It is even possible that examiner A's unconscious bias may have shifted the data for the control group. Thus, these points need to be addressed appropriately and explained sufficiently for the manuscript to be accepted

Author Response

Response to Reviewer 1 Comments

Thank you very much again for the review of our manuscript (jcm-1803434). The comments of the review were constructive and have been used to revise and improve the manuscript. We enclose a revised version of our paper titled “The radiographic risk factors predicting occurrence of idiopathic carpal tunnel syndrome in simple wrist x-ray” by BeomSu Han, KiHong Kim and KyuJin Kim as a submission to Journal of Clinical Medicine. We highlighted the edits made to the original version of the manuscript with red color in the revised manuscript. The following is an itemized account of the changes in the manuscript made in response to the comments.

Point 1: Page 3, line 104.

Based on your description, since rater A measured the radiographic factors unblinded, his/her data should have been biased.

Ideally, please include another rater (rater C) who is blinded and have him/her measure the data. Then, please remove the data measured by rater A, add the data measured by rater C, and analyze them again.

If you cannot do it for some reason, please present the data obtained by each rater and the agreement/consistency between the two raters' results. Those might allow you to indicate that the measurement values obtained by rater A did not favorably affect the analysis results.

Response 1: Thank you for your valuable comment. We included another rater (rater C) who is blinded and have him measure the data. Then, we removed the data measured by rater A, added the data measured by rater C. We analyzed it again.

Point 2: Page 4, line 133 (Table 1).

In your study, in which many women were included, the RLA of the control group was -3.9 degrees, but in the following past references A and B, the RLA is presented as 9.3 and 3.3 degrees, respectively. In particular, reference B shows that RLA is significantly greater in women. Unlike your study, the studies in references A and B involved almost equal numbers of men and women; thus, I must say that there is a rather large difference between their results and yours. If these were the results in the control group in your study, it is possible that you would not have obtained a significant difference.

Then, looking at your methods section, you state that you randomly included patients as a control group who were performed x-ray for screening purposes. What disease or reason did these patients have their wrist x-rays done for screening purposes? Were they really appropriate as a control group? Please show the evidence that the data in the control group were appropriate enough for the research. Also, please discuss and provide the reason for the differences in results from previous papers

Response 2: Thank you for your valuable comment. We agree with you. So, we added another rater (rater C) and rater C measured the radiographic angle again in blinded state. As a result, the RLA of the control group was -1.1 degrees. Although it is still negative value unlike reference A and B (9.3 and 3.3 degrees), the difference has decreased a lot (-3.9 degrees -> -1.1 degrees). Also, we found other references which dealing with normal range of carpal alignment angle.

Larsen et al. mentioned the mean angle of RLA as -1.02 degrees (DOI : 10.1016/s0363-5023(10)80156-x). Lee et al. mentioned the mean angle of RLA as +1.2 degrees (DOI : 10.1016/j.jhsa.2018.01.003). Considering the above two references, the average RLA value of our study does not show much difference.                                 

More women are involved in our study. To match the sex ratio of the control and comparison groups, more women in control group is inevitable considering that idiopathic CTS is more prevalent in women.

Thank you for your comment of methods section. We have corrected the methods section. In our center, all patients who visit with symptoms such as pain in the wrist must take X-rays of the opposite wrist to compare with the symptomatic area. Medical records were reviewed that there were no symptoms in the opposite wrist and we measured radiographic angle using X-rays of the opposite wrist. 

Reviewer 2 Report

The study has some interesting points, because it treats a topic with a great impact in the common orthopedics practice.The purpose is clear and respected: to consider RLA and LARD during surgery for distal radius fractures to prevent future CTS development.This is an essential step not only to restore correct radiocarpal function but also to prevent the onset of CTS. The information provided is sufficient.The review is based on meaningful documents with good impact.

Author Response

Thank you very much again for the review of our manuscript (jcm-1803434). The comments of the review were constructive and have been used to revise and improve the manuscript. We enclose a revised version of our paper titled “The radiographic risk factors predicting occurrence of idiopathic carpal tunnel syndrome in simple wrist x-ray” by BeomSu Han, KiHong Kim and KyuJin Kim as a submission to Journal of Clinical Medicine. We highlighted the edits made to the original version of the manuscript with red color in the revised manuscript. 

Round 2

Reviewer 1 Report

The manuscript was revised better and is in good condition now.

I recommend that it be accepted for publication.

Just for the information.

Reference A is the following

https://pubmed.ncbi.nlm.nih.gov/12413964/

and, Reference B is the following

https://pubmed.ncbi.nlm.nih.gov/29299502/